# Periodontitis Prevalence, Severity, and Risk Factors: A Comparison of the AAP/CDC Case Definition and the EFP/AAP Classification

**DOI:** 10.3390/ijerph18073459

**Published:** 2021-03-26

**Authors:** Meliha Germen, Ulku Baser, Cagdas Caglar Lacin, Erhan Fıratlı, Halim İşsever, Funda Yalcin

**Affiliations:** 1Periodontology Department, Faculty of Dentistry, University of Istanbul, 34134 Istanbul, Turkey; melihagermen@hotmail.com (M.G.); cagdascaglarlacin@gmail.com (C.C.L.); erhanfiratli@gmail.com (E.F.); fundaylcn@yahoo.com (F.Y.); 2Department of Public Health, Faculty of Medical, University of Istanbul, 34104 Istanbul, Turkey; hissever@istanbul.edu.tr

**Keywords:** periodontitis, prevalence, risk factors, smoking, age, systemic disease, diabetes, classification

## Abstract

Background: This cross-sectional study evaluated the utility of the 2018 European Federation of Periodontology/American Academy of Periodontology (EFP/AAP) classifications of epidemiological studies in terms of periodontitis severity, prevalence and associated risk factors and the 2012 American Academy of Periodontology/Centers for Disease Control and Prevention (AAP/CDC) case definitions. Methods: We included 488 participants aged 35–74 years. Measurements were recorded at six sites per tooth by two qualified examiners. The evaluated parameters included pocket depth (PD), clinical attachment loss (CAL) and bleeding on probing (BOP). Periodontitis prevalence and severity were reported using the 2018 EFP/AAP classification and the AAP/CDC case definitions. The data were stratified by recognized risk factors (age, diabetes and smoking status). Results: The 2018 EFP/AAP classification indicated that all patients suffered from periodontitis. When CAL served as the main criterion, the frequency of patients with severe (Stages III–IV) periodontitis was 54%. When the AAP/CDC case definitions were applied, the prevalence of periodontitis was 61.9% and that of severe periodontitis 16.8%. Age was the most significant risk factor, regardless of the chosen case definition. Conclusion: It is essential to employ a globalized standard case definition when monitoring periodontitis and associated risk factors.

## 1. Introduction

Periodontitis is common, reflecting complex interactions between pathogenic periodontal microbiota and the host immune response, modulated by environmental and genetic factors [1,2]. A risk factor may be an environmental exposure, a behavioral trait or an inherent characteristic associated with a disease [2]. Many studies have shown that periodontal disease incidence is affected by age, smoking, diabetes and socioeconomic status [3,4,5,6]. Risk factors play roles in both the initiation and progression of periodontal disease [7]. A rigorous analysis of evidence supposedly supporting the roles of various risk factors in terms of periodontitis prevalence and severity is very important when diagnosing and treating periodontal disease. The periodontitis prevalence varies among both developed and developing countries [8,9,10,11,12,13,14]. The historical lack of a standard periodontitis definition facilitating surveillance precludes any meaningful comparisons of findings in terms of variations in socioeconomic status, ethnicity or exposure to risk factors. Furthermore, the lack of consistency in terms of both case definitions and clinical examination protocols, and differences in sample demographics, limit comparisons between our present findings and the literature data [15,16]. The European Association of Dental Public Health (2010) recommends the combined use of the Clinical Attachment Loss (CAL), Probing Depth (PD) and Bleeding on Probing (BOP) tests; these three key variables should be assessed in all future epidemiological studies on periodontal diseases [17]. Various case definitions have been employed in previous studies, including the definition of the American Academy of Periodontology/Centers for Disease Control and Prevention (AAP/CDC) [18,19,20]. The National Health and Nutrition Examination Survey (NHANES) survey of 2009–2010 was the first to apply the AAP/CDC case definition published by Page and Eke in 2007 [19,21]. Today, a new periodontitis classification scheme has been adopted; this features multidimensional staging and grading. Staging is dependent upon disease severity at presentation and the complexity of the required management [22]. Here, we assess periodontitis prevalence and severity using the 2018 European Federation of Periodontology/American Academy of Periodontology (EFP/AAP) classification and the AAP/CDC case definitions. We evaluate the associations between prevalence levels and the periodontitis severity and risk factors in an adult subpopulation.

## 2. Materials and Methods

### 2.1. Study Population

We included 488 participants aged 35–74 years. We collected data from September 2012 to November 2013 in Istanbul, Turkey. The survey center was a public health, primary care facility selected by searching the statistical data of the Ministry of Health of Turkey. Individuals who required antibiotics after routine periodontal procedures were excluded. Participants were randomly selected from daily attendees. All were informed about the nature of the research, and all signed consent forms prior to study entry. The study protocol was approved by the Ethics Committee of the Istanbul University Faculty of Medicine (approval no. 2013/1071).

### 2.2. Examination Protocol and Measurements

Prior to clinical examination, all participants were asked to complete a history-taking questionnaire, and to record data on age, gender, smoking status (non-smoker, current smoker or former smoker) and systemic health status (healthy, diabetic or other systemic disease). Smoking status was defined as follows: non-smokers, participants who have never smoked, current smokers, participants who currently smoke more than 10 cigarettes per day, former smokers, participants who have quit smoking more than 1 year ago. All clinical examinations were performed by two calibrated examiners. The intraclass correlation coefficient for the site-level PD ranged from 0.86 to 0.89, and the interclass correlation coefficients were 0.93 for the first and 0.97 for the second examinations. Full-mouth measurements were recorded at six sites per tooth (distobuccal, buccal, mesiobuccal, distolingual, lingual and mesiolingual) (excluding the third molars); all measurements were obtained under similar optimal lighting. The evaluated parameters included the PD, CAL and BOP. The CAL was the distance from the cemento-enamel junction (CEJ) to the free gingival margin (FGM) added to the distance from the FGM to the bottom of the pocket/sulcus. A periodontal probe (Hu-Friedy, Chicago, IL, USA) with 1-, 2-, 3-, 5-, 7-, 8-, 9- and 10-mm gradations was positioned parallel to the long axes of all teeth to facilitate clinical measurements. Measurements were made in mm and rounded to the next mm. The BOP was scored after all teeth were probed.

### 2.3. Definition of Periodontitis

#### 2.3.1. AAP/CDC Periodontitis Case Definition

The prevalence of periodontitis was recorded using the suggested AAP/CDC case definitions [19,20]. Severe periodontitis was defined as the presence of two or more interproximal sites with AL values ≥ 6 mm (not for the same teeth) and one or more interproximal site(s) with PDs ≥ 5 mm. Moderate periodontitis was defined as two or more interproximal sites with clinical ALs ≥ 4 mm (not for the same teeth) or two or more interproximal sites with PDs ≥ 5 mm, also not for the same teeth. Mild periodontitis was defined as two or more interproximal sites with ALs ≥ 3 mm and two or more interproximal sites with PDs ≥ 4 mm (not for the same teeth) or one site with a PD ≥ 5 mm. The final periodontitis score was the sum of the severe, moderate and mild scores [19,20].

#### 2.3.2. EFP/AAP Classification

Periodontal disease was diagnosed with reference to the classification proposed at the 2018 World Workshop on the Classification of Periodontal and Peri-Implant Diseases and Conditions, as follows: the inter-dental CAL of two non-adjacent teeth, or the buccal or oral CAL, was ≥3 mm, with pocketing >3 mm [23]. Periodontitis severity staging was calculated accordingly: for each tooth, the CAL of the most severe site was recorded; a CAL of 1–2 mm was defined as Stage I, of 3–4 mm as Stage II and of ≥ 5 mm as Stages III–IV. The number of missing teeth was not considered, because of the lack of sufficient data. Periodontitis staging reflected the complexity of management. Stage I or II patients were reclassified as Stages III–IV if the maximum PD was ≥6 mm. 

The prevalence of periodontitis was also reported using two CAL cutoff points, ranging from 3 to 7 mm. The extent of periodontitis was the summed percentages of sites and teeth for which the CALs attained the same cutoff points.

### 2.4. Statistical Analysis

We performed a power analysis to estimate the minimum acceptable sample size, thus affording a prevalence outcome of interest of 50% and an acceptable error of 6%. All data were analyzed using the SPSS statistical software package (ver. 22.0; SPSS, Chicago, IL, USA). The chi-squared test was used to compare individuals with and without periodontal disease and groups of varying disease severity, as defined above.

## 3. Results

### 3.1. Periodontitis Prevalence and Severity According to the AAP/CDC Case Definition

From 2012 to 2013, the prevalence of periodontitis in the 488 participants aged 35–74 years was 61.9% by the CDC/AAP case definition. The total prevalence was 61.9% by this definition. The prevalence rates of mild, moderate and severe periodontitis were 17.8%, 27.3% and 16.8%, respectively. The prevalence of periodontitis ranged from 57.5% in adults 35–44 years of age to 85.7% in those 65–74 years of age. The prevalence of severe periodontitis was significantly lower in adults aged 35–44 years than in those aged 45–64 and 65–74 years (Table 1).

### 3.2. Periodontitis Prevalence and Severity by the EFP/AAP Classification

According to the 2018 classification, all subjects exhibited periodontitis. When the CAL served as the main criterion, the severity staging was as follows: Stage I, 41% (200/488); Stage II, 25% (123/488); and Stage III–IV, 34% (165/488) (Table 1). Significant differences were evident between groups stratified by age and smoking status (both *p* < 0.001) (Table 2).

When staging was graded in terms of the complexity of management, 21/488 subjects (6%) were classified as Stage I, 194/488 (40%) as Stage II and 263/488 (54%) as Stages III–IV (Table 2). Periodontitis staging in terms of complexity by age group, smoking status and systemic health is shown in Table 3. A significant difference was evident between groups stratified by age (*p* < 0.001). Smoking seemed to accompany greater disease severity, but the association was not significant (*p* = 0.062). 

### 3.3. Periodontitis Prevalence and Extent by the Clinical Attachment Level Threshold Values

Table 4 shows the prevalence and extent of periodontitis by the numbers and percentages of sites and teeth with CALs of 3–7 mm. CALs ≥ 4 mm were recorded for 162 individuals (74%), for 804 teeth (15.5%) and at 1521 sites (5%) in those aged 35–44 years. CALs ≥ 7 mm were observed in 31 (14.2) individuals for 56 teeth (1.1%) and at 107 sites (0.3%) in the same age group (Table 4).

## 4. Discussion

We compared the 2012 AAP/CDC periodontitis case definition and the 2018 EFP/AAP periodontal disease classification in terms of patient characteristics, risk factors and disease severity and extent. We enrolled adults aged 35–74 years, all of whom had periodontitis by the 2018 EFP/AAP classification. The total periodontitis prevalence by the 2012 AAP/CDC definition was 61.9%. The difference between 61% and 100% is attributable to the use of different definitions when examining the same subpopulation. In the new classification, periodontitis is defined as CAL at two non-adjacent teeth [22,23]. Even though the case definition [23] does not stipulate a threshold (which might avoid misclassification), any numerical CAL value must be defined epidemiologically. Compared to the AAP/CDC definition [19,20] (a CAL ≥ 3 mm), Stage I represents early periodontitis, which is difficult to distinguish from advanced gingivitis. Thus, the decrease in the CAL threshold of the new classification increased the prevalence of periodontitis.

We determined periodontitis severity using CAL as the principal criterion: Stage I: 41%, Stage II: 25%, Stages III–IV: 34% (Table 2). When the maximum PD value was added when determining complexity, the distribution of periodontitis stages changed dramatically (Table 3). Only a few epidemiological studies have used the new classification [12,24,25,26,27,28]. However, the epidemiological surveys did not clearly explain how complexity was used to determine periodontitis staging. Here, we list the severity and complexity parameters in two separate tables to show how the staging changes depending on the parameters employed. Although the CAL is the principal severity criterion of the new classification, the complexity of management requires evaluation of furcation, pocket depth and mobility. If complexity management is to be used to stage disease in epidemiological studies, the evaluations to be included must be clarified (one or all of mobility, pocket depth and a furcation defect). 

Using the AAP/CDC case definition, the prevalence rates of mild, moderate and severe periodontitis were 17.8%, 27.3% and 16.8%, respectively. Compared to the prevalence afforded by the new classification, severe periodontitis is underestimated when using the AAP/CDC classification. The AAP/CDC case definition threshold is two or more interproximal sites with AL ≥ 6 mm and at least one interproximal site(s) with a PD ≥ 5 mm, but the new classification defines severe periodontitis (Stage III–IV) as a site with an AL ≥ 5 mm. Table 4 shows how the CAL prevalence changes by the chosen cutoff point (the threshold value). Therefore, the numbers of patients with ALs ≥ 6 mm (AAP/CDC) compared to ≥ 5 mm (EFP/AAP) explains the observed severity differences. 

In the 2018 EFP/AAP classification, the number of missing teeth is one factor relevant to staging. Stage I or II periodontitis defined by clinical attachment loss is redefined as Stage III–IV if one or more teeth have been lost to periodontal disease. Therefore, it is critical to clarify (via history-taking) why teeth have been lost, especially in cross-sectional studies lacking long-term follow-up. As this work is cross-sectional in nature, we restaged a patient only if self-reported tooth loss was attributable to periodontal disease.

When data were stratified by risk factors (age, smoking and diabetes), we found that the prevalence of periodontitis was considerably higher in older age groups, in agreement with the literature [6,29]. Both the severity and extent of CAL significantly increased with age. Furthermore, the highest prevalence of periodontitis was evident in subjects aged 65–74 years, and the prevalence of severe periodontitis (defined by the AAP/CDC and AAP/EFP classifications) was significantly higher among those aged 45–64 and 65–74 years than 35–44 years. Regardless of the chosen definition, age exerted the strongest influence on periodontal disease, in agreement with other studies [6,16,30]. A recent work found that the relationship between periodontitis and diabetes was limited to only severe periodontitis and uncontrolled diabetes. The 2018 classification recommends that the HbA1c value should be used to sub-grade disease. As a limitation, in the present study, all disease was self-reported and glycated hemoglobin levels were not measured; we were thus unable to explore any possible causal relationship [16,25]. Eke et al. stated that periodontitis was significantly more prevalent in current and former smokers than non-smokers [16]. Furthermore, periodontitis was most common in current smokers, and smoking was strongly associated with severe periodontitis. We found that smoking was indeed a risk factor reflected by the new classification, in which CAL is the recognized severity criterion (Table 2). 

Unfortunately, no uniform criteria for periodontal disease have yet been established. Costa et al. [31] assessed the effects of five different definitions on the prevalence and extent of periodontitis, and found that the rate varied greatly, from 13.8% to 65.3% in terms of prevalence and from 9.7% to 55.6% in terms of extent. When defining periodontitis, the combined use of the CAL and PD was suggested to be more reliable; CAL measurements optimally estimate damage to the periodontium, whereas PD measurements best predict attachment loss [17]. The situation is further confused by the variations in the thresholds used to define cases, regardless of the criteria applied [17]. We present our data using the CDC/AAP case definition, the 2018 EFP/AAP classification and the CAL; all are used to monitor periodontitis. Given the different CAL and PD thresholds used in the various epidemiological studies, we here report the CAL prevalence and extent at different cutoff points ranging from 3 to 7 mm; we emphasize that the methodology affects the results. 

Benigeri et al. [32] found that only 8.5% of adults had at least one tooth with a 6 mm or deeper periodontal pocket after probing at two sites, but this figure rose 2.5-fold (to 21.4%) when all teeth were probed. In our study, 74 of every 100 subjects exhibited pockets of depth 4 mm or greater, but this decreased to 15.5% on a tooth basis and to 5% on a site basis in those aged 35–44 years. Circumferential full-mouth measurements are essential; otherwise, disease will be underestimated. 

Methodological differences are a major problem for comparison between studies, especially when preparing meta-analyses or reviews. The most common of these methodological difficulties involves the use of different definitions of periodontal diseases. Systematic reviews of the literature are often hindered by the lack of a uniform definition of periodontitis. In addition, depending on the definition or measurement of periodontitis, different results can be obtained when evaluating the relationships of periodontitis with other systemic health conditions. It is not possible to compare two studies of periodontitis and outcomes or risks if the two studies are not uniform in their definition of periodontitis [33].

We are aware that it is not possible to compare a case definition to a multidimensional classification that facilitates a comprehensive assessment of risk and complexity factors. We discuss the possible utility of the new classification in terms of epidemiological surveys, and we compare the current and past definitions of periodontitis. 

## 5. Conclusions

In conclusion, use of the new classification to assess periodontitis prevalence, severity and risk factors when conducting epidemiological surveys may not be totally reliable. The new classification allows comprehensive assessment. However, it is too vague for use in population surveys. Within the limitations of our study, we found that age maximally influenced periodontitis status; any association of other risk factors varied by the survey methodology and the parameters measured. Constant updating is essential to align any classification scheme to the current understanding of periodontal disease. The combined use of CAL and PD thresholds would ensure the comparability of studies on different populations in various countries and between the same populations over time.

## Figures and Tables

**Table 1 ijerph-18-03459-t001:** Prevalence of periodontitis by the characteristics of the CDC/AAP (American Academy of Periodontology/Centers for Disease Control and Prevention) case definition.

	Mild PD (*n* = 87)	Moderate PD (*n* = 133)	Severe PD (*n* = 82)	Total PD (*n* = 302)	*p*-Value
Age (years)					
35–44	44 (20.1%)	64 (29.2%)	18 (8.2%)	126 (57.5%)	0.001 *
45–64	36 (14.5%)	63 (25.4%)	59 (23.8%)	158 (63.7%)	
65–74	7 (33.3%)	6 (28.6%)	5 (23.8%)	18 (85.7%)	
Smoking status					
Non-smoker	45 (16.1%)	79 (28.2%)	49 (17.5%)	173 (61.8%)	0.871
Current smoker	28 (19.7%)	35 (24.6%)	22 (15.5%)	85 (59.8%)	
Former smoker	14 (21.2%)	19 (28.8%)	11 (16.7%)	44 (66.7%)	
Systemic diseasestatus					
None	48 (16.1%)	90 (30.2%)	48 (16.1%)	186 (62.4%)	0.072
Diabetes	15 (28.3%)	13 (24.5%)	12 (22.6%)	40 (75.3%)	
Other	24 (17.8%)	30 (21.9%)	22 (16.1%)	76 (55.5%)	

Pearson’s chi-squared test: * *p* < 0.05. PD: Periodontitis.

**Table 2 ijerph-18-03459-t002:** Severity of periodontitis by the characteristics of the EFP/AAP (European Federation of Periodontology/American Academy of Periodontology) 2018 classification.

	Stage I (*n* = 200)	Stage II (*n* = 123)	Stage III–IV (*n* = 165)	Total PD (*n* = 488)	*p*-Value
Age (years)					
35–44	122 (55.7%)	54 (24.6%)	43 (19.7%)	219 (100%)	0.0001 *
45–64	76 (30.6%)	63 (25.4%)	109 (43.9%)	248 (100%)	
65–74	2 (9.5%)	6 (28.5%)	13 (61.9%)	21 (100%)	
Smoking status					
Non-smoker	121 (43.3%)	61 (21.7%)	98 (35%)	280 (100%)	0.031 *
Current smoker	62 (43.7%)	37 (26%)	43 (30.3%)	142 (100%)	
Former smoker	17 (25.8%)	25 (38%)	24 (36.4%)	66 (100%)	
Systemic disease status					
None	122 (40.9%)	71 (23.8%)	105 (35.2%)	298 (100%)	0.099
Diabetes	14 (26.4%)	19 (35.8%)	20 (37.7%)	53 (100%)	
Other	64 (46.7%)	33 (24.1%)	40 (29.2%)	137 (100%)	

Pearson’s chi-square test * *p* < 0.05. PD: Periodontitis. Severity was measured with reference to the loss of attachment cutoff points.

**Table 3 ijerph-18-03459-t003:** The complexity of periodontitis management staging by the various characteristics of the EFP/AAP 2018 classification.

	Stage I (*n* = 31)	Stage II (*n* = 194)	Stage III–IV (*n* = 263)	Total PD (*n* = 488)	*p*-Value
Age (years)					
35–44	23 (10.5%)	124 (56.6%)	72 (32.9%)	219 (100.0%)	0.0001 *
45–64	8 (3.2%)	69 (27.8%)	171 (69.0%)	248 (100.0%)	
65–74	0 (0.0%)	1 (4.8%)	20 (95.2%)	21 (100.0%)	
Smoking Status					
Non-smoker	22 (7.9%)	100 (35.7%)	158 (56.4%)	280 (100.0%)	0.062
Current smoker	9 (6.3%)	64 (45.1%)	69 (48.6%)	142 (100.0%)	
Former smoker	0 (0.0%)	30 (45.5%)	36 (54.5%)	66 (100.0%)	
Systemic disease					
None	20 (6.7%)	116 (38.9%)	162 (54.4%)	298 (100.0%)	0.583
Diabetes	2 (3.8%)	18 (34.0%)	33 (62.3%)	53 (100.0%)	
Other	9 (6.6%)	60 (43.8%)	68 (49.6%)	137 (100.0%)	

Pearson’s chi-square test: * *p* < 0.05. The complexity of management was measured by addition of the maximum probing depth after severity staging via loss of attachment. PD: Periodontitis.

**Table 4 ijerph-18-03459-t004:** The prevalence and extent of teeth and sites meeting the threshold values of clinical attachment loss by age (years).

CAL.	35–44 (*n* = 219)		45–64 (*n* = 248)		65–74 (*n* = 21)		Total (*n* = 488)		*p*-Value
	*n*	%	*n*	%	N	%	*n*	%	
Prevalence (individual) (*n* = 488)									
≥3 mm	196	89.5	240	96.8	21	100	457	93.6	0.003 **
≥4 mm	162	74	214	86.3	20	95.2	396	81.1	0.001 **
≥5 mm	71	32.4	171	69	20	95.2	262	53.7	0.001 **
≥6 mm	51	23.3	121	48.8	14	66.7	186	38.1	0.001 **
≥7 mm	31	14.2	79	31.9	11	52.4	121	24.8	0.001 **
Extent (no. of teeth) (*n* = 10,517)									
≥3 mm	1.614	31.1	2462	49.4	195	58.2	4.271	40.6	0.001 **
≥4 mm	804	15.5	1538	30.9	129	38.5	2.471	23.5	0.001 **
≥5 mm	315	6.1	855	17.2	76	22.7	1.246	11.8	0.001 **
≥6 mm	136	2.6	485	9.7	41	12.2	662	6.3	0.001 **
≥7 mm	56	1.1	274	5.5	21	6.3	351	3.3	0.001 **
Extent (no. of sites) (*n* = 63,102)									
≥3 mm	3.502	11.2	6932	23.2	529	26.3	10963	17.4	0.001 **
≥4 mm	1.521	4.9	3.842	12.8	329	16.4	5.692	9	0.001 **
≥5 mm	577	1.9	1.923	6.4	152	7.6	2.652	4.2	0.001 **
≥6 mm	239	0.8	1.026	3.4	73	3.6	1.338	2.1	0.001 **
≥7 mm	107	0.3	534	1.8	29	1.4	670	1.1	0.001 **

Pearson’s chi-square test: ** *p* < 0.01.

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
