# Peer review of "Periodontitis Prevalence, Severity, and Risk Factors: A Comparison of the AAP/CDC Case Definition and the EFP/AAP Classification"

_ijerph, 2021, doi:10.3390/ijerph18073459_

Round 1

Reviewer 1 Report

This paper seeks to look at how varying definitions of periodontitis lead to varying prevalence reports of periodontitis within populations.

What the authors did not include in their discussion that might be worth considering is: 

This work is also important because it is difficult to study the association of periodontitis with other systemic health conditions without a uniform definition. IE, of recent memory, many have been trying to tie Alzheimer's risk to periodontitis, but without a clear definition of periodontitis this is difficult. Additionally, systematic reviews of the literature are often hindered by a lack of uniform definition of periodontitis. We can't compare two studies of periodontitis and outcome or risk if those two studies aren't uniform on how they defined the 'periodontitis' population.

Also see: 

Manau, C., A. Echeverria, A. Agueda, A. Guerrero, and J. J. Echeverria. 2008. Periodontal disease definition may determine the association between periodontitis and pregnancy outcomes. J Clin Periodontol 35: 385-397. (Conflict of interest note: This reviewer is not affiliated with this paper) 

I would consider this ready for publication if 1) a good English-language reviewer corrected the few and very minor errors in grammar and 2) the above points were at least briefly considered in the discussion. 

This is a very important topic that many aren't talking about. 

Author Response

Response

  • We have added a paragraph in accordance with the reviewer’s comment as shown below and, have included the recommended article in the References.

Methodological differences are a major problem for comparison between studies, especially when preparing meta-analyses or reviews. The most common of these methodological difficulties involves the use of different definitions of periodontal diseases. Systematic reviews of the literature are often hindered by the lack of a uniform definition of periodontitis. In addition, depending on the definition or measurement of periodontitis, different results can be obtained when evaluating the relationships of periodontitis with other systemic health conditions. It is not possible to compare two studies of periodontitis and outcomes or risks if the two studies are not uniform in their definition of periodontitis [33].

  • The English in the manuscript has been checked by professional editing service.

Reviewer 2 Report

Dear Authors,

I have carefully read Your paper entitled “Periodontitis Prevalence, Severity and Association with risk Factors. Comparison of AAP/CDC Case Definition and EFP/AAP Classification".

I think this work still needs some remarks before its publication.

Paragraph 2.2 Examination protocol and measurements
Regarding the definition of the identified risk factors, some clarifications are required in order to better understand this aspect of your analysis:
-    As per the smoking status, you identified three categories (non-smokers, current smokers, former smokers). This characterization requires the exact definition for each of the subgroups (e.g., how do you define a former smoker?).
-    You seem to not have accounted for the severity of the smoking habit. Do you think a clinical quantification of cigarette smoking (i.e. pack-year score) could provide a better inside of the subjects’ smoking habit, and therefore a better stratification of the sample?
-    Similarly, diabetes as a risk factor is poorly defined. Which kind of diabetes are you evaluating (e.g. type I, II, etc.)? Was it compensated or not? 
-    Additionally, what’s the subjects’ history of dental treatments? Did they undergo a dental hygiene prior to your clinical evaluation? If yes, how much time before your enrollment?

Author Response

-   As per the smoking status, you identified three categories (non-smokers, current smokers, former smokers). This characterization requires the exact definition for each of the subgroups (e.g., how do you define a former smoker?)

Response

In the original questionnaire of the study, we grouped smoking status as shown below. However, at the end of the study, we reduced the number of groups to three because there were very few responses in FS1 and OS.

Non-smokers (NS): never smoked

Occasional smokers (OS; < 10 cigarettes a day)

Moderate smokers (MS; 10–19 cigarettes a day)

Heavy smokers (HS; > 20 cigarettes a day)

Former smokers (FS1 > 1 year since cessation)

Former smokers (FS2 < 1 year since cessation)

We have added the smoking habit methodology into the manuscript as follows.

Smoking status was defined as follows: non-smokers, participants who have never smoked, current smokers, participants who currently smoke more than 10 cigarettes per day, former smokers, participants who have quit smoking more than 1 year ago.

-   You seem to not have accounted for the severity of the smoking habit. Do you think a clinical quantification of cigarette smoking (i.e. pack-year score) could provide a better inside of the subjects’ smoking habit, and therefore a better stratification of the sample?

Response

As noted by the reviewer, we did not obtain pack-year data that could give a better idea of the subjects’ smoking habits. Other studies (Eke et al. 2020, Graetz et al. 2019, Iao et al. 2020) examining the prevalence of periodontal diseases and risk factors according to the definitions of the diseases generally evaluated smoking status in three groups, as in our study.

-   Similarly, diabetes as a risk factor is poorly defined. Which kind of diabetes are you evaluating (e.g. type I, II, etc.)? Was it compensated or not?

Response

We did not determine the glycated haemoglobin levels of participants who reported having diabetes. This has been mentioned in the Discussion section as a limitation of our study as shown below.

As a limitation, in the present study, all disease was self-reported and glycated haemoglobin levels were not measured; we were thus unable to explore any possible causal relationship [16,25].

-   Additionally, what’s the subjects’ history of dental treatments? Did they undergo a dental hygiene prior to your clinical evaluation? If yes, how much time before your enrollment?

Response

A questionnaire was used to obtain data on smoking habit (non-smoker, current smoker or former smoker), educational level of the individual or his/her parents (last school level completed), oral hygiene habits (frequency of tooth brushing), frequency of dental visits (regular or irregular), marital status (married, separated, divorced, widow/widower or never married) and monthly income (based on last annual income per person).

However, we did not include this information in the manuscript. The goal of the present study was not to determine the prevalence of periodontitis in a specific population, but rather to highlight the differences in results that may be obtained in epidemiological studies depending on the definitions or measurements of periodontitis.

In addition, we did not provide any oral hygiene instruction or treatment as this was a model of an epidemiological study.
